# Nuclear Imaging for the Diagnosis of Cardiac Amyloidosis in 2021

**DOI:** 10.3390/diagnostics11060996

**Published:** 2021-05-30

**Authors:** Weijia Li, Dipan Uppal, Yu Chiang Wang, Xiaobo Xu, Damianos G. Kokkinidis, Mark I. Travin, James M. Tauras

**Affiliations:** 1Department of Medicine, Jacobi Medical Center, Albert Einstein College of Medicine, 1400 Pelham Parkway South, Bronx, NY 10461, USA; uppald@nychhc.org (D.U.); wangy19@nychhc.org (Y.C.W.); xux7@nychhc.org (X.X.); 2Section of Cardiovascular Medicine, Yale University School of Medicine, 333 Cedar Street, New Haven, CT 06510, USA; damianos.kokkinidis@yale.edu; 3Department of Radiology, Division of Nuclear Medicine, Montefiore Medical Center, Albert Einstein College of Medicine, 111 East 210th Street, Bronx, NY 10467, USA; mtravin@montefiore.org; 4Department of Medicine, Division of Cardiology, Montefiore Medical Center, Albert Einstein College of Medicine, 111 East 210th Street, Bronx, NY 10467, USA; jtauras@montefiore.org

**Keywords:** cardiac amyloidosis, cardiac scintigraphy, positron emission tomography

## Abstract

Cardiac amyloidosis is caused by the deposition of misfolded protein fibrils into the extracellular space of the heart. The diagnosis of cardiac amyloidosis remains challenging because of the heterogeneous manifestations of the disease. There are many different types of amyloidosis with light-chain (AL) amyloidosis and transthyretin (ATTR) amyloidosis being the most common types of cardiac amyloidosis. Endomyocardial biopsy is considered the gold standard for diagnosing cardiac amyloidosis and differentiating amyloid subtypes, but its use is limited because of the invasive nature of the procedure, with risks for complications and the need for specialized training and centers to perform the procedure. Radionuclide cardiac imaging has recently become the most commonly performed test for the diagnosis of ATTR amyloidosis but is of limited value for the diagnosis of AL amyloidosis. Positron emission tomography has been increasingly used for the diagnosis of cardiac amyloidosis and its applications are expected to expand in the future. Imaging protocols are under refinement to achieve better quantification of the disease burden and prediction of prognosis.

## 1. Introduction

Systemic amyloidosis is a multisystem disorder characterized by the formation and deposition of mis-folded protein fibrils which can result in multi-organ failure and death [1,2]. This condition is associated with significant disease burden with increasing incidence and prevalence worldwide over the past decades [3,4,5]. Studies have shown that at least 20 out of one million UK residents are estimated to have systemic amyloidosis with 65% being light-chain (AL) amyloidosis. The prevalence of wild type ATTR amyloidosis is estimated to be 10–25% in people over the age of 80 [6,7,8,9].

Cardiac amyloidosis is defined as a group of disorders that involve the deposition of amyloid protein in the cardiac tissue, leading to myocardial dysfunction [10]. Due to the increasing awareness of the disease, improved life expectancy, and advancements in diagnostic pathways, cardiac amyloidosis is currently diagnosed more frequently than in the past, with AL and ATTR amyloidosis being the most common types [3]. One population-based study focusing on Medicare beneficiaries in the United States revealed that between 2000 and 2012 the prevalence of cardiac amyloidosis increased from 8 to 17 cases per 100,000 person-year and the incidence increased from 18 to 55 cases per 100,000 person-year [11]. It is estimated that by 2050 there will be almost 25 million cases of wild-type ATTR globally [12,13].

AL amyloidosis is caused by the deposition of fragments of immunoglobulin and is associated with plasmacyte disorders such as multiple myeloma [14]. On the other hand, ATTR amyloidosis results from misfolded transthyretin, which normally functions as a transporter for thyroid hormone and retinol-binding protein [15]. ATTR amyloidosis can be further divided into wild-type which develops as people age and a hereditary form caused by inherited mutations [16,17].

AL cardiac amyloidosis and ATTR cardiac amyloidosis are clinically distinctive diseases and the diagnosis and differentiation are of vital importance, mainly because of the drastically different treatment strategies [18]. For example, the development of tafamidis, which acts on the rate-limiting step of the amyloidogenic process by binding to transthyretin, stabilizing the tetramer and slowing the dissociation into monomers, has revolutionized the therapeutic landscape for ATTR amyloidosis. On the other hand, the management for AL cardiac amyloidosis focuses on the control of the underlying plasmacyte disorder such as autologous stem cell transplantation and bortezomib-based treatment [19]. Thus, the early and accurate diagnosis of cardiac amyloidosis is essential [20]. Endomyocardial biopsy (EMB) is considered the gold standard for the diagnosis of cardiac amyloidosis when combined with mass spectrometry to ascertain specific sub-types [21,22]. However, given the heterogeneity of cardiac amyloidosis, EMB may fail to diagnose the disease. Additionally, it is an invasive procedure by nature and thus can lead to serious complications such as right ventricle perforation, cardiac tamponade, and even death [23,24].

Nuclear imaging as an alternative diagnostic approach has become more popular among clinicians who suspect cardiac amyloidosis in their patients. In addition, compared to other imaging modalities such as echocardiogram and cardiac magnetic resonance imaging (CMR), nuclear imaging is the only available non-invasive method that can accurately distinguish ATTR amyloidosis from AL cardiac amyloidosis within the appropriate clinical settings [25]. Nuclear imaging has been viewed as an essential part of the diagnosis of cardiac amyloidosis and has been included in the multi-society consensus statement published in 2019 [26,27]. With this review article, we aim to summarize and discuss the current landscape of nuclear imaging for the diagnosis of cardiac amyloidosis, recent advancements and expected future changes in the field and how nuclear imaging can help cardiovascular clinicians diagnose cardiac amyloidosis.

## 2. Imaging Techniques and Radiotracers

Nuclear imaging for cardiac amyloidosis includes cardiac scintigraphy which adopts radiotracers from bone scintigraphy and the technique of Positron Emission Tomography (PET) using targeted tracers for amyloid specific proteins (Figure 1). Nuclear imaging can offer direct visualization of disease activity and semi-quantification of the amyloid burden by calculating the ratio between concentration of the radiotracer in a specific volume of tissue and the concentration if the radiotracers are uniformly distributed. The ratio is also known as Standardized Uptake Value (SUV) [28]. Retention index (RI), calculated to assess the retention of radiotracers in myocardium over certain time interval, can be another important method measuring amyloid deposition quantitatively in PET imaging [29,30,31].

## 3. Cardiac Scintigraphy

Technetium (Tc)-labeled radiotracers from phosphate derivatives which are common bone scan agents have been investigated for the diagnosis of amyloidosis since the 1970s [32]. Cardiac scintigraphy utilizing those radiotracers has become more popular and widely used in clinical practice to assist the diagnosis of cardiac amyloidosis. A large-sized multi-center study including 1217 patients with suspected cardiac amyloidosis supported that cardiac scintigraphy with ^99m^Tc-diphosphono-1,2-propanodicarboxylic acid (^99m^Tc-DPD), ^99m^Tc-pyrophosphate (^99m^Tc-PYP), or ^99m^Tc-hydroxymethylene diphosphonate (^99m^Tc-HMDP) had a 100% specificity and positive predictive value for ATTR cardiac amyloidosis if there are combined findings of grade 2 or 3 myocardial radiotracer uptake on cardiac scintigraphy and the absence of monoclonal protein in serum or urine [33]. Systematic review also confirmed the accuracy of scintigraphy diagnosing ATTR cardiac amyloidosis with both sensitivity and specificity above 90% [34]. Although it remains unclear how those bone-seeking agents can differentiate the types of amyloidosis, theories have postulated that the higher calcium containing compounds in ATTR cardiac amyloidosis, the unique characteristics of amyloidogenic fibrils, and the more indolent clinical course of ATTR amyloidosis allowing larger amount of amyloid protein to accumulate before the onset of symptoms might play a role [35]. However, the results of cardiac scintigraphy can be affected by multiple factors such as rib fracture and valvular/annular calcifications [36]. Single-Photon Emission Computed Tomography (SPECT) technique can add a three-dimensional visualization to planar scintigraphy as well as more detailed and accurate assessment of radiotracer uptake in the myocardium wall as opposed to the blood pool [37]. Currently, ^99m^Tc-DPD and ^99m^Tc-PYP are the two most commonly used and studied radiotracers for the diagnosis of ATTR cardiac amyloidosis.

## 4. ^99m^Tc-3,3-diphosphono-1,2-propanodicarboxylic Acid (^99m^Tc-DPD) Scintigraphy 

^99m^Tc-DPD scintigraphy is usually performed as a whole-body planar imaging three hours after the injection of ^99m^Tc-DPD radiotracer which can be followed by SPECT and non-contrast CT as adjuncts. ^99m^Tc-DPD scintigraphy has high diagnostic accuracy for ATTR amyloidosis, especially when using the Perugini visual score. According to the Perugini visual score, the degree of radiotracer uptake is visually graded by comparing the radiotracer activity in the heart with its activity in the bones. Grade 0 means no cardiac uptake of radiotracer; grade 1 means that cardiac uptake is mild and less than skeletal uptake; grade 2 means that cardiac uptake is moderate and equals skeletal uptake; grade 3 means cardiac uptake is high and stronger than skeletal uptake. Grade 2 and above are considered as positive scan [38]. However, this protocol can lead to false positive results in patients with AL cardiac amyloidosis [33]. Therefore, AL cardiac amyloidosis needs to be ruled out first with serum free light chains, serum protein electrophoresis, and urine protein electrophoresis before interpreting the results of cardiac scintigraphy. As the visual scoring system highly depends on reader expertise, it performed poorly when assessing the degree of amyloid burden [39]. In addition, Perugini visual score has not been found to have any prognostic significance in the overall survival for patients with cardiac amyloidosis [40].

Researchers have attempted to increase the diagnostic accuracy of ^99m^Tc-DPD scintigraphy and quantitatively assess amyloid burden by calculating the ratio between retention of radiotracer in the heart and retention of radiotracer in other body parts. Heart/whole-body ratio (H/WB), heart/pelvis ratio and heart/contralateral lung ratios (H/CL) are commonly used in clinical studies [41]. ATTR amyloidosis was found to have a higher H/WB ratio than AL amyloidosis. A study from Australia which enrolled biopsy-proven AL and ATTR cardiac amyloidosis has proposed a cut-off of H/WB ratio >0.091 with sensitivity of 92% and specificity of 88% for the diagnosis of ATTR amyloidosis [42]. Additionally, an increasing H/WB ratio has been shown to correlate with major adverse cardiac events in patients with hereditary ATTR cardiac amyloidosis [43]. Interestingly, researchers also found that ^99m^Tc-DPD scintigraphy might have a role in the diagnosis of extracardiac AL amyloidosis when cardiac uptake is absent [44].

SPECT/CT has been developed to assist quantification in ^99m^Tc-DPD Scintigraphy by acquiring peak Standard Uptake Values (SUVs) in the myocardium and offering three-dimensional assessment [45]. The cardiac peak SUV can be further normalized with the peak SUV on the bone or soft tissue as SUV retention index [46]. Studies have shown that cardiac SUV and SUV retention index are correlated well with Perugini visual scores and a peak SUV cut-off of 3.1 can separate patients with Perugini grade 2 and 3 clearly from those with Perugini grade 0 and 1 [47]. A recent study has found that the amyloid load in ^99m^Tc-DPD SPECT/CT has correlated well with strain values in echocardiography and biomarkers such as troponin and NT-proBNP (B-type Natriuretic Peptide) [48]. However, SPECT/CT was still unable to differentiate between patients with Perugini grade 2 and 3, which suggests that quantification of amyloid burden by ^99m^Tc-DPD SPECT/CT needs further improvement [39].

## 5. ^99m^Tc-Pyrophosphate (^99m^Tc-PYP) Scintigraphy 

Although promising, ^99m^Tc-DPD is not approved for use by the Food and Drug Administration (FDA) in the United States. Hence, ^99m^Tc-pyrophosphate (^99m^Tc-PYP) is the only FDA-approved radiotracer in the US to diagnose cardiac amyloidosis [49]. Clinicians usually obtain anterior, lateral and left anterior oblique planar views as well as SPECT imaging following injection of ^99m^Tc-PYP [50]. The degree of myocardial tracer uptake is graded using the semi-quantitative Perugini visual score and quantitative analysis by obtaining radiotracer activity within a region of interest (ROI) drawn over the heart corrected and its activity in the contralateral side of ROI to calculate a heart-to-contralateral (H/CL) ratio [51]. Unlike the 3-h protocol which is required in ^99m^Tc-DPD scintigraphy, it has been found that a 1-h protocol in ^99m^Tc-PYP imaging is comparable to the 3-h protocol for the diagnosis of ATTR cardiac amyloidosis. This translates to a 98% sensitivity and a 96% specificity of planar imaging and SPECT, identical between the 1-h and 3-h protocols [52]. The 1-h protocol reduces cost and time without compromising the diagnostic accuracy of the test and thus it is widely used.

Bokhari et al. found that subjects with ATTR cardiac amyloidosis had a significantly higher cardiac visual score (*p* < 0.0001) as well as higher H/CL ratio (*p* < 0.00001) than AL amyloidosis and they concluded that using a H/CL ratio of ≥ 1.5, which is consistent with intensely diffused myocardial tracer retention, had a 97% sensitivity and 100% specificity (*p* < 0.0001) for identifying ATTR cardiac amyloidosis [35]. In a multicenter study which enrolled 171 participants, ^99m^Tc-PYP scan demonstrated an overall 91% sensitivity and 92% specificity for detecting ATTR cardiac amyloidosis with area under the curve of 0.960 (95% CI, 0.930–0.981) [53]. It has also been noted that an H/CL ratio ≥ 1.6 predicts lower 5-year survival compared with group of patients with an H/CL ratio ≤ 1.6 (log-rank *p* = 0.02) [53]. Despite the high accuracy of the ^99m^Tc-PYP scan visual score and H/CL ratio, the addition of SPECT is still necessary to rule out misclassified cases and distinguish myocardial activity from blood pool uptake [54]. A recent study showed that combining ^99m^Tc-PYP and Thallium (Tl)-201 may improve diagnostic accuracy of both visual differentiation and H/CL semi-quantification for ATTR amyloidosis [55]. In addition, an integrated approach of utilizing both high sensitivity cardiac troponin T and ^99m^Tc-PYP scintigraphy can significantly increase diagnostic yield of wild-type ATTR cardiac amyloidosis [56]. A series of studies of ^99m^Tc-DPD and ^99m^Tc-PYP scintigraphy published from 2020 to 2021 are listed in Table 1.

## 6. Positron Emission Tomography (PET)

Positron emission tomography (PET) scanning is another imaging modality which can help diagnose cardiac amyloidosis [63]. PET imaging offers higher spatial resolution secondary to the decay of positrons and more accurate quantification of amyloid burden by using direct amyloid-binding radioactive tracers [64]. ^11^C-Pittsburgh B (^11^C-PiB) and ^18^F-labelled agents (such as ^18^F-florbetapir and ^18^F-florbetaben) are the two most common classes of radioactive tracers used for this purpose [65]. The tracers were originally developed to bind beta amyloid in the brain of patients with Alzheimer disease but it was reported later that they might have utility in diagnosing cardiac amyloidosis as well [66]. Higher cardiac uptake of both ^11^C-PiB and ^18^F-labelled agents was constantly observed in both AL cardiac amyloidosis and ATTR cardiac amyloidosis, compared to controls in pilot studies [29,60,67,68,69,70]. Additionally, the radiotracer activities of both ^11^C-PiB and ^18^F-labelled agents have been found to be higher in AL cardiac amyloidosis than ATTR amyloidosis. A meta-analysis which combined the results of three pilot PET studies demonstrated that AL amyloidosis has significantly higher radiotracer activities than ATTR amyloidosis and thus PET imaging carries the potential to differentiate between AL and ATTR amyloidosis [71]. Overall, PET imaging for the diagnosis of cardiac amyloidosis is still in the early stages but future development of this technique is anticipated.

## 7. ^11^C-Pittsburgh Compound B PET Imaging

^11^C-Pittsburgh compound B (^11^C-PiB) PET imaging is a well-established technique for detecting β-amyloid in Alzheimer disease [72]. PiB, thioflavin-T, is an amyloid binding dye, and is theoretically able to bind to amyloid fibrils of any type, including amyloid fibrils in the myocardium [73]. A Swedish study included 10 patients with systemic amyloidosis (7 AL, 2 hereditary ATTR, 1 wild-type ATTR) and cardiac involvement, and the results showed increased myocardial ^11^C-PiB uptake in all the patients 15–25 min after injection of ^11^C-PiB. On the other hand, increased uptake was not seen in the five patients of the control group [29]. A Korean prospective study, which included 22 amyloidosis patients (15 with and 7 without cardiac involvement) and 10 normal controls, calculated the SUV and found significantly higher values in patients with cardiac amyloidosis than the control group (median 3.9 (range 1.7 to 19.9) vs. 1.0 (range 0.8 to 1.2), *p* < 0.001) [67].

^11^C-PiB PET imaging shows promise for identifying specific types of cardiac amyloidosis, especially for the AL subtype. A dual-center study showed that ^11^C-PiB PET imaging had 100% diagnostic accuracy of AL amyloidosis and that the uptake was significantly higher in AL cardiac amyloidosis compared to ATTR cardiac amyloidosis [58]. Researchers from Korea compared ^11^C-PiB PET imaging with endomyocardial biopsy in patients with chemotherapy-naive AL cardiac amyloidosis and they found that the degree of the ^11^C-PiB uptake on PET image was significantly higher in patients with cardiac amyloidosis and it corresponded well with the extent of amyloid deposition on the biopsy specimens. Patients with higher ^11^C-PiB uptake had high risks of composite adverse clinical outcomes including death, requiring heart transplantation and acute decompensated heart failure [59].

A study conducted in Japan revealed that ^99m^Tc-PYP scintigraphy and ^11^C-PiB PET imaging can complement each other. In this study, the combination of positive ^11^C-PiB PET and negative ^99m^Tc-PYP was observed in all AL cardiac amyloidosis and early onset V30M hereditary ATTR cardiac amyloidosis, while the combination of positive ^99m^Tc-PYP and negative ^11^C-PiB PET was consistent in all wild-type ATTR cardiac amyloidosis, as well as the late-onset V30M and non-V30M hereditary ATTR cardiac amyloidosis [57]. However, ^11^C-PiB PET imaging is limited by its short half-life of 20 min and the requirement of an onsite cyclotron for its production [74].

## 8. ^18^F-Labelled Agents PET Imaging

^18^F-labelled PET imaging is another promising imaging technique utilizing fluoride labelled radiotracers, primarily ^18^F-florbetapir and ^18^F-florbetaben. ^18^F-florbetapir and ^18^F-florbetaben are both FDA-approved radioactive tracers for Alzheimer’s disease with half-lives more than 100 min, which is a potential advantage over ^11^C-PiB PET for use in clinical practice. [75,76]. A pilot study in 2014 which enrolled 14 subjects (9 subjects with definite cardiac amyloidosis and 5 control subjects without amyloidosis) found that myocardial retention of ^18^F-florbetapir was higher in amyloid subjects, especially in patients with AL cardiac amyloidosis [77]. An autoradiography study using myocardial autopsy sections yielded similar results, showing that ^18^F-florbetapir uptake was higher in amyloid samples versus controls but also found higher uptake in the AL groups compared to the ATTR samples [78]. Another pilot study in 2016 indicated that ^18^F-florbetaben PET imaging can accurately diagnose and differentiate between cardiac amyloidosis and hypertensive heart disease [69]. The percentage of myocardial ^18^F-florbetaben retention was found to be an independent determinant of myocardial dysfunction in cardiac amyloidosis [69]. A pilot study in 2019 involving 22 subjects (5 proven and 17 with clinical suspicion of cardiac amyloidosis) revealed that ^18^F-florbetaben-PET could further distinguish between the underlying amyloid types with higher retention in patients with AL amyloidosis than ATTR amyloidosis [79]. A recently published prospective study showed that delayed ^18^F-florbetaben cardiac uptake may distinguish AL cardiac amyloidosis from ATTR amyloidosis given higher mean SUV in patients with AL amyloidosis which was sustained over the whole acquisition period [60]. In addition, amyloid-directed PET can be used to assess therapy response. It has been shown that amyloid burden on PET after treatment with anti-inflammatory (AA), anti-myeloma (AL) and TTR-stabilizing (ATTR) therapies correlated well with changes in performance status and serological markers [79]. A series of most recent studies for ^11^C-PiB and ^18^F-labelled agents PET imaging can be found in Table 1.

Researchers have combined fluoride PET imaging with MRI in patients with cardiac amyloidosis to improve diagnostic accuracy of ATTR amyloidosis [61]. However, while PET imaging can distinguish between cardiac amyloidosis and controls, particularly when using quantitative analysis, it seems to be less sensitive when diagnosing cardiac amyloidosis than the more established nuclear medicine studies with ^99m^Tc-PYP or ^99m^Tc-DPD [62,80]. The comparison between cardiac scintigraphy and PET imaging is summarized in Table 2.

Our review highlights the importance of nuclear imaging for the diagnosis of cardiac amyloidosis with most updated clinical evidence and covers the most common radiotracers in this field. We not only elaborate on cardiac scintigraphy which is the more established nuclear imaging modality for cardiac amyloidosis, but also include the most recent evidence regarding PET imaging. However, we do acknowledge that other radiotracers, for example ^99m^Tc-HMDP, are present and may play a role in the diagnosis of cardiac amyloidosis. In addition, we are unable to find large-sized clinical studies to compare between those radiotracers mentioned in this review.

## 9. Conclusions

Cardiac scintigraphy with SPECT is the current standard of care for diagnosing patients with ATTR cardiac amyloidosis. However, PET imaging is another promising, non-invasive option for the diagnosis of cardiac amyloidosis and may help distinguish between AL amyloidosis and ATTR amyloidosis. The potential benefit of PET-based radiotracers includes better sensitivity for AL cardiac amyloidosis diagnosis and assessment of response to treatment. Future studies are anticipated.

## Figures and Tables

**Figure 1 diagnostics-11-00996-f001:**
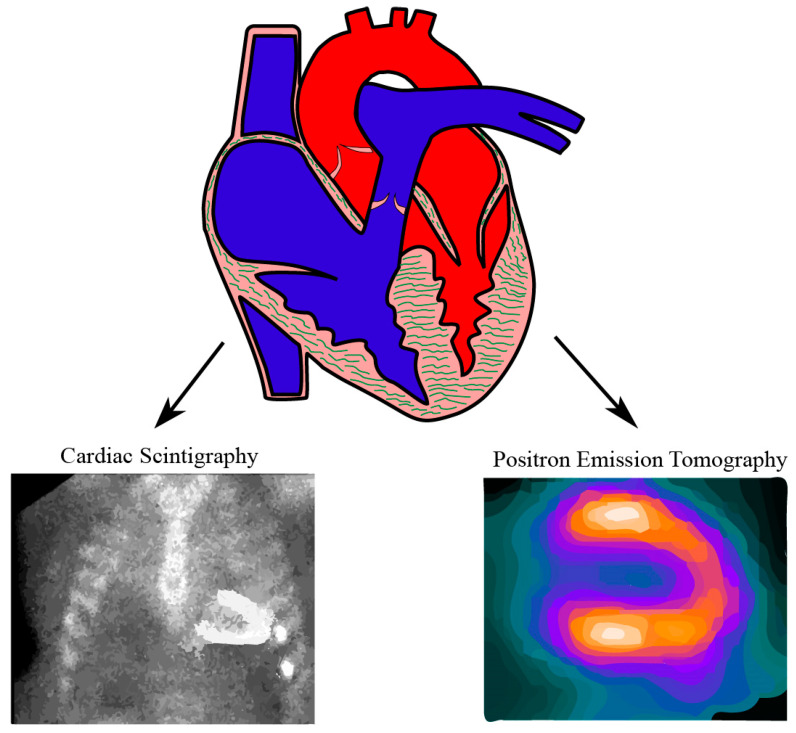
Illustration of nuclear imaging modalities for cardiac amyloidosis.

**Table 1 diagnostics-11-00996-t001:** Summary of a series of published studies of nuclear imaging in cardiac amyloidosis from 2020 to 2021.

First Author	Publication Year	Radiotracer	Method	Results
Caobelli et al. [39]	2020	^99m^Tc-DPD	Retrospective single-center study including 13 patients with 8 ATTR cardiac amyloidosis and 5 not.	Myocardial SUV_max_ and SUV_peak_ showed strong correlation with Perugini score but a great degree of overlap between patients in Perugini score 2 and 3.
Scully et al. [46]	2020	^99m^Tc-DPD	Single-center, retrospective study of 100 DPD scan (40 were Perugini grade 0, 12 were grade 1, 41 were grade 2, and 7 were grade 3).	SUV retention index which is calculated as: ((Cardiac SUV_peak_/Vertebral SUV_peak_) × paraspinal muscle SUV_peak_) increased across all Perugini grades. Cardiac SUV_peak_ and SUV retention index had excellent diagnostic accuracy with the area under the curve being 0.999.
Wollenweber et al. [47]	2020	^99m^Tc-DPD	32 patients with bioptically-proven or suspected cardiac ATTR amyloidosis received a DPD total body bone scan with additional SPECT/CT.	Patients with Perugini grade 2 and 3 can be clearly separated from those with Perugini grade 0 and 1 with a SUV_peak_ cut-off of 3.1.
Löfbacka et al. [48]	2020	^99m^Tc-DPD	48 patients with genetically-verified hereditary ATTR cardiac amyloidosis and positive ^99m^Tc-DPD SPECT/CT were assessed manually for amyloid burden.	Statistically significant correlation between DPD uptake and all echocardiographic strain parameters in all regions, as well as the biomarkers of troponin and logarithmic NT-proBNP.
Masri et al. [52]	2020	^99m^Tc-PYP	233 patients with suspected ATTR cardiac amyloidosis underwent planar and SPECT imaging at 1 and 3 hours with a positive scan considered as visual grades ≥ 2 and heart to contralateral ratios ≥ 1.5	1-hour and 3-hour protocols have identical SPECT results. Planar imaging at 1 hour had 98% sensitivity and 96% specificity.
Asif et al. [54]	2020	^99m^Tc-PYP	^99m^Tc-PYP scintigraphy was performed including 1-hour planar imaging assessing visual score as well as H/CL ratio and SPECT	Visual score had a diagnostic accuracy of 98% for ATTR cardiac amyloidosis but addition of H/CL ratio reduced the accuracy. SPECT is necessary to perform to prevent misdiagnoses.
Tamarappoo et al. [55]	2020	^99m^Tc-PYP/Tl-201	Dual isotope of ^99m^Tc-PYP/Tl-201 SPECT was performed in 112 patients suspicious of cardiac amyloidosis (39 ATTR, 26 AL, 47 no amyloidosis) and compared with single isotope. H/CL ratio was calculated.	Interobserver agreement of visual assessment was better with dual-isotope SPECT. Area under the curve for ATTR cardiac amyloidosis by visual assessment and H/CL ratio were higher with dual-isotope SPECT than single-isotope SPECT.
Ochi et al. [56]	2020	^99m^Tc-PYP	39 patients with wild-type ATTR cardiac amyloidosis with 8 patients in group A who were diagnosed before the introduction of hs-cTnT and ^99m^Tc-PYP scintigraphy and 31 patients in group B who were diagnosed after the introduction of the two tools.	Increased diagnostic yield in patients who used the combined approach using hs-cTnT and ^99m^Tc-PYP scintigraphy.
Takasone et al. [57]	2020	^99m^Tc-PYP, ^11^C-PiB	17 patients with AL cardiac amyloidosis, 22 patients with hereditary ATTR cardiac amyloidosis, and 8 patients with wild-type ATTR cardiac amyloidosis underwent both ^11^C-PiB PET imaging and ^99m^Tc-PYP scintigraphy.	All patients with cardiac amyloidosis are detectable by ^99m^Tc-PYP or ^11^C-PiB PET imaging. The combination of positive ^11^C-PiB PET and negative ^99m^Tc-PYP was observed in all AL cardiac amyloidosis and early onset V30M hereditary ATTR cardiac amyloidosis, while the combination of positive ^99m^Tc-PYP and negative ^11^C-PiB PET was consistent in all wild-type ATTR cardiac amyloidosis, as well as the late-onset V30M and non-V30M hereditary ATTR cardiac amyloidosis.
Rosengren et al. [58]	2020	^11^C-PiB	A dual-center study included 51 subjects with 36 patients with known cardiac amyloidosis and increased wall thickness (15 AL, 21 ATTR) and 15 control patients. All the subjects underwent ^11^C-PiB PET imaging and echocardiography.	High diagnostic accuracy of both visual inspection and semi-quantitative methods of ^11^C-PiB PET imaging to distinguish cardiac amyloidosis from controls. The uptake of ^11^C-PiB was significantly higher in AL cardiac amyloidosis than ATTR cardiac amyloidosis.
Lee et al. [59]	2020	^11^C-PiB	41 chemotherapy-naïve AL cardiac amyloidosis patients were enrolled. Myocardial uptake of ^11^C-PiB on PET was compared with endomyocardial biopsy for quantification of amyloid deposit.	The degree of myocardial ^11^C-PiB uptake is significantly higher in patients with cardiac amyloidosis and higher degrees of uptake was associated with lowest survival from death, heart transplantation and acute decompensated heart failure.
Genovesi et al. [60]	2021	^18^F-florbetaben	40 patients with biopsy-proven diagnoses of cardiac amyloidosis (20 AL amyloidosis, 20 ATTR amyloidosis) and 20 patients with non-cardiac amyloidosis pathology.	Patients with AL amyloidosis have higher mean SUV, heart-to-background uptake ratio, and molecular volume than ATTR amyloidosis and patients with non-cardiac amyloidosis.
Andrews et al. [61]	2020	^18^F-fluoride	A prospective multicenter study included 53 patients (10 ATTR and 8 AL cardiac amyloidosis, 13 controls and 22 with aortic stenosis). All patients were scanned by ^18^F-fluoride PET/MRI. SUV and tissue-to-background ratio (TBR_mean_) were obtained in the septum and areas of late gadolinium enhancement.	TBR_mean_ values are higher in ATTR amyloidosis than controls and those with AL amyloidosis. A TBR_mean_ threshold >1.14 in areas of late gadolinium enhancement has 100% sensitivity and 100% specificity for ATTR amyloidosis compared to AL amyloidosis.
Zhang et al. [62]	2020	^18^F-sodium fluoride and ^99m^Tc-PYP	12 subjects with ATTR cardiac amyloidosis and 5 controls underwent ^18^F-sodium fluoride and ^99m^Tc-PYP-SPECT/CT.	Visual assessment of ^18^F-sodium fluoride PET/CT had a sensitivity of 25% for ATTR cardiac amyloidosis when compared with 100% sensitivity in ^99m^Tc-PYP-SPECT/CT.

^99m^Tc-DPD: ^99m^Tc-3,3-diphosphono-1,2-propanodicarboxylic acid; ^99m^Tc-PYP: ^99m^Tc-pyrophosphate; ^11^C-PiB: ^11^C-Pittsburgh compound B.

**Table 2 diagnostics-11-00996-t002:** Comparison between the four most-studied nuclear imaging techniques for cardiac amyloidosis.

Imaging Technique	Radiotracer Component	Radiotracer Analog	Radiotracers Original Application	Amyloidosis Type	Advantage	Disadvantage
^99m^Tc-DPD Scintigraphy	^99m^Tc-3,3-diphosphono-1,2-propanodicarboxylic acid	Phosphate	Bone scintigraphy	ATTR amyloidosis >> AL amyloidosis.	High diagnostic accuracy for ATTR when combined with SPECT and the absence of a monoclonal protein in serum or urine.	Limited on accurate quantification of amyloid burden.
^99m^Tc-PYP Scintigraphy	^99m^Tc-Pyrophosphate					
^11^C-PiB PET imaging	N-methyl-[^11^C]2-(4′-methylaminophenyl)-6-hydroxybenzothiazole	Thioflavin-T	Brain imaging in Alzheimer dementia.	AL amyloidosis > ATTR amyloidosis.	Detect both AL and ATTR amyloidosis, ability to detect early disease, short study session. Can complement ^99m^Tc-PYP Scintigraphy.	Requirement of onsite cyclotron for generation; high synthesis cost with 20-min half-life.
^18^F-labelled agents PET imaging	^18^F-florbetapir, ^18^F-florbetaben^18^F-NaF	Stilbene			Can diagnose both AL amyloidosis and ATTR amyloidosis. Allows for early detection of cardiac amyloidosis, aid in therapy response assessment.	Lack of large-sized studies to confirm its efficacy.

^99m^Tc-DPD: ^99m^Tc-3,3-diphosphono-1,2-propanodicarboxylic acid; ^99m^Tc-PYP: ^99m^Tc-pyrophosphate; ^11^C-PiB: ^11^C-Pittsburgh compound B.

## Data Availability

Not applicable.

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
