# Peer review of "Nuclear Imaging for the Diagnosis of Cardiac Amyloidosis in 2021"

_diagnostics, 2021, doi:10.3390/diagnostics11060996_

Round 1
Reviewer 1 Report
Comments on the review manuscript "Nuclear Imaging for the Diagnosis of Cardiac Amyloidosis in 2021" by Weijia and colleagues.
In general this is a well written manuscript, some uncertainties need however to be clarified.
Imaging Techniques and Radiotracers: "
Nuclear imaging can offer direct visualization of disease activity and semi-quantification of the amyloid burden by calculating the ratio between concentration of the radiotracer in a specific volume of tissue and the concentration if the radiotracers are uniformly distributed. The ratio is also known as Standardized Uptake Value (SUV)" Also retention index (RI) is an important quantitative PET measurement. This should be added.
Figure 1: the bone scan is of moderate quality.
Cardiac Scintigraphy (page 3): also add the systymatic review of the diagnostic value of bone imaging in cardiac amyloidosis (Treglia et al, PMID: 29687207).
Cardiac Scintigraphy when the Perugini score is 1-2, and the diagnose is unclear, AL should be excluded and serum & urine tests are needed to rule out light-chain amyloidosis.
Cardiac Scintigraphy: extra-cardiac findings should be reported (positive organs (liver/spleen/lymph nodes), soft tissue).
Cardiac Scintigraphy:: mention the potential pitfalls (or small table) of bone scanning in CA. See also the paper of Garcia-Pavia et al, Eur Heart J. 2021 Apr 21;42(16):1554-1568. doi: 10.1093/eurheartj/ehab072. Look for Table 4: Possible false positives and false negatives of bisphosphonate scintigraphy for detecting transthyretin cardiac amyloidosis.
Table 1: several studies are lacking, adapt the title into: Summary of a serie of published studies of..
11C-PIB, should be written as 11C-PiB (officially even as: [11C]PiB)
Reviewer 2 Report
I consider that this narrative review presents in an original manner the current evidence regarding the importance of nuclear imaging in the diagnosis of patients with cardiac amyloidosis. The article is well structured and the data presented in Table 2 summarize very well the advantages and disadvantages of the four most-studied nuclear imaging techniques. I consider that the article may be published in this journal, after some minor revisions:
- The authors should mention the limits and especially the novelty of their review.
- Considering the recent position statement of the ESC Working Group on Myocardial and Pericardial Diseases regarding the diagnosis and treatment of cardiac amyloidosis, I suggest including some data about the importance of imaging modalities in the diagnostic algorithm of patients with cardiac amyloidosis.
Reviewer 3 Report
The manuscript is a comprehensive analysis of nuclear imaging in cardiac amyloidosis.
I suggest to indicate the acronysm of the tracers under the tables.
I suggest also to perform a language review: some parts are not easily to read, in particular where many acronysm are used into the main body.
Round 2
Reviewer 1 Report
My comments are addressed well, but just have a look at figure 1, it looks a bit "compressed".
Author Response
We appreciate the reviewer's comment. We have revised Figure 1 as suggested.